# Evaluation of a Passive Upper Limb Exoskeleton in Healthcare Workers during a Surgical Instrument Cleaning Task

**DOI:** 10.3390/ijerph20043153

**Published:** 2023-02-10

**Authors:** Bastien Arnoux, Anaïs Farr, Vincent Boccara, Nicolas Vignais

**Affiliations:** 1CIAMS, Université Paris-Saclay, 91405 Orsay, France; 2CIAMS, Université d’Orléans, 45067 Orléans, France; 3LIMSI CNRS, Université Paris Sud XI, CEDEX, 91403 Orsay, France

**Keywords:** passive exoskeleton, musculoskeletal disorders, electromyography, healthcare workers

## Abstract

(1) Background: Healthcare workers are highly affected by work-related musculoskeletal disorders, particularly in the lower back, neck and shoulders, as their occupational tasks expose them to biomechanical constraints. One solution to prevent these musculoskeletal disorders may be the use of a passive exoskeleton as it aims to reduce muscle solicitation. However, few studies have been carried out directly in this field to assess the impact of the use of a passive upper limb exoskeleton on this population. (2) Methods: Seven healthcare workers, equipped with electromyographic sensors, performed a tool cleaning task with and without a passive upper limb exoskeleton (Hapo MS, Ergosanté Technologie, France). Six muscles of the upper limbs were analysed, i.e., anterior deltoid, biceps brachii, pectoralis major, latissimus dorsi, triceps brachii and longissimus thoracis. A subjective analysis of the usability of the equipment, the perception of effort and discomfort, was also carried out using the System Usability Scale and the Borg scale. (3) Results: The longissimus thoracis was the most used muscle during this task. We observed a significant decrease in the muscular solicitation of the anterior deltoid and latissimus dorsi when wearing the exoskeleton. Other muscles were not significantly impacted by the device. (4) Conclusions: the passive exoskeleton used in this study allowed the reduction in muscular load on the anterior deltoid and latissimus dorsi without negative effects on other muscles. Other field studies with exoskeletons are now necessary, particularly in hospitals, to increase our knowledge and improve the acceptability of this system for the prevention of musculoskeletal disorders.

## 1. Introduction

Work-related musculoskeletal disorders (MSDs) are occupational injuries that affect the musculoskeletal system of the human body, including bones, spinal discs, tendons, joints, ligaments, cartilage, nerves and blood vessels [1]. This disease impacts a large part of the working population. For example, MSDs were the cause of 88% of occupational illnesses in France in 2020 [2], with 30% of them related to the shoulders [3]. More than 100 million people in Europe are affected by MSDs [4], which are also responsible for 60% of permanent work disabilities, and 50% of all work absences in Europe. MSDs have an impact on both workers’ performances and their personal lives [5], and they represent a significant economic cost. Indeed, in certain European nations, MSDs represented 40% of the cost of workers’ compensation, which lowers the gross domestic product of each state by 1% to 2% [5]. Some employees are more at risk of developing MSDs, such as healthcare workers [6]. This prevalence of MSD appearance in hospital workers is due to the biomechanical loads caused by physical tasks such as repetitive movements or lifting tasks, e.g., lifting and transferring patients, washing patients, cleaning rooms, lifting heavy surgical instruments [7,8]. In this population, MSDs mainly affect the lower back, neck and shoulders [7,9,10].

The use of exoskeletons is increasingly seen as a solution to prevent the appearance of MSDs [11]. An exoskeleton can be defined “as a wearable, external mechanical structure that enhances the power of a person” [12]. Two types of exoskeleton can be distinguished: active and passive exoskeletons [13]. An active exoskeleton is composed of actuators (hydraulic, pneumatic muscles, electric motors or other) that increase the power of the human. Unlike the active exoskeleton, the passive exoskeleton does not use actuators, but springs or dampers that store the energy created by human movement and then release it to help the human support a posture or movement. Thus, passive exoskeletons may be more easily used in industry to reduce MSDs that widely affect workers. Indeed, according to the literature, the use of passive exoskeletons decreases the EMG amplitude of the arm and shoulder muscles, notably the anterior deltoid, medialis deltoid, upper trapezius and biceps brachii [3,14,15,16,17,18,19,20,21,22,23,24,25,26,27,28]). There is also a subsequent decrease in fatigue when the subject wears the exoskeleton compared to situations without the exoskeleton [29,30,31,32].

Although there is no current scientific consensus, exoskeletons may also have negative effects on the user. According to some studies, the use of exoskeletons could be accompanied by an increase in the muscular activity of other muscles, in particular lumbar erector spinae and triceps brachii [14,16,22,27]. However, other investigations did not highlight this increase in EMG amplitude for the triceps brachii [3] and lumbar erector spinae [18,20]. These adverse outcomes may be related to the fact that studies have been carried out either into a laboratory or directly in the field. Of all the studies with passive upper limb exoskeletons, only nine have been conducted in the field [19,20,23,26,29,33,34,35,36]. Results of these studies are mostly in agreement with those carried out into a laboratory environment, with a fatigue reduction and a decrease of muscle solicitation for shoulder muscles, while wearing the exoskeleton. However, field studies using exoskeletons are essential to better target the needs of certain professions when implementing MSDs prevention strategies [26].

Thus, healthcare workers have been rarely concerned with physical assistance dedicated to the prevention of MSDs. Only few studies have been carried out on the impact of upper limb exoskeletons, particularly under field conditions into a hospital environment [31]. However, although this latter study was performed in a hospital environment, it focused on the surgeon during an operation. Thus, the aim of this study is therefore to analyse the influence of wearing an upper limb exoskeleton on healthcare workers during a task with high repetition, which reflects their biomechanical solicitation, i.e., a surgical tool cleaning task. The primary hypothesis is that the upper limb exoskeleton may decrease the activity of muscles involved in this task. The secondary hypothesis suggests that using an upper limb exoskeleton may increase the muscular activity of back muscles, as showed in the literature.

## 2. Materials and Methods

### 2.1. Participants

For this study, seven healthcare workers (one woman and six men) from the surgical department of a hospital in France were recruited. Their mean age, height and weight were 36.63 ± 8.78 years, 178.88 ± 5.34 cm and 85.13 ± 8.91 kg, respectively. To participate in this study, subjects should not have history of upper limb injuries, and had to be of legal age. All participants were informed about the procedure prior to the experiment, and read and signed their informed consent before the beginning of the experiment. The protocol and data acquisition process were validated by a local Research Ethics Committee (University of Paris-Saclay, 2022-451).

### 2.2. Materials

#### 2.2.1. Exoskeleton

For this study, a passive exoskeleton dedicated to the upper limbs was used (Hapo MS^™^, ErgoSanté Technologie, Anduze, France). This light passive exoskeleton (1.3 kg) has been designed to provide physical assistance to the upper limbs. More precisely, this exoskeleton allows the arms to be relieved over an amplitude of 0 to 135° in the sagittal plane, and 0° to 180° in the frontal plane.

This exoskeleton is composed of a harness that is worn on the back with an attachment that encompasses the elbow and is connected to the harness via a spring (Figure 1). The stiffness of the springs can be 4 kg for light assistance or 6 kg for heavy assistance, and this can be modified to suit the type of work and the user’s morphology. The 4 kg (or 6 kg) assistance allows the extremity of the spring to be held at 90°, with respect to the shoulder level on the harness, with two weights of 2 kg (or 3 kg) at each extremity. In the current study, an assistance of 4 kg has been used given the fact that most movements were below 90° of arm flexion. The exoskeleton was adjusted to each participant through several adjustable attachments (waist belt, horizontal starting point of the tube integrating the spring at back level, length of the spring through telescopic tube, chest belt).

#### 2.2.2. EMG

Six electromyographic sensors (Biometrics, DataLITE^®^, Biometrics Ltd., Newport, UK) were used to measure the muscle activation of the subject with a sample frequency of 1000 Hz. Three muscles involved in arm flexion were selected, i.e., anterior deltoid, biceps brachii, and pectoralis major), as well as two antagonist muscles, i.e., latissimus dorsi and triceps brachii. The Longissimus thoracis was also selected to analyse a potential negative effect of the exoskeleton on back muscles, in accordance with the literature. The EMGs were placed according to the SENIAM recommendations [38]. Before placing the electrodes, the area was shaved and cleaned. Then, the electrodes were placed on the muscle of interest in the direction of the muscle fibres.

In parallel with the EMG recording, movements were videotaped using a Sony Handycam HDR-XR260^®^ camera (Sony, Tokyo, Japan), in order to identify the movements made by the subject and to synchronise the EMG signal with the video recordings.

#### 2.2.3. Questionnaires

Perceived task effort, perceived discomfort while wearing the exoskeleton and usability of the exoskeleton were also assessed. To this aim, two questionnaires were created using the Borg scale ranging from 0 (nothing at all) to 10 (extremely strong) [39]. The questions aimed at evaluating the effort and discomfort perceived during the task were as follows:-From 0 to 10, how would you rate the physical effort of the task?-From 0 to 10, how would you rate the discomfort felt during the task?

The usability questionnaire was based on the System Usability Scale (SUS) [40]. The SUS is a scale used to measure the usability of a product or service. It consists of a series of 10 statements, scaled on a 5-item scale ranging from one (strongly disagree) to five (strongly agree). A score of 100 is then calculated, and if the score is above 70, then the usability of the system is considered acceptable [41].

### 2.3. Procedure

The protocol consisted of performing a common task in the profession of heathlcare workers that was also identified as a traumatic task for the shoulders [42], the cleaning of surgical tools.

First, the subject was taken to a separate room where the experimenter was able to place the six EMG electrodes on the subject’s upper limbs and trunk. The subject’s maximum voluntary contractions (MVC) was then collected over three trials [43]. After this preparation step, the subject performed a tool cleaning task, in one of the two conditions, with and without exoskeleton, in a randomised order (Figure 2). The task lasted between 5 and 20 min to complete, depending on the quantity and nature of the instruments that needed to be cleaned. If the subject performed another tool cleaning task immediately afterwards, we did not register a new MVC and we performed the second condition directly (with or without the exoskeleton depending on the condition that was already done). If it was not possible to record successively the second condition, he or she was completely unequipped. In that case, the second condition was recorded later in the day and the MVC was collected a second time before this condition.

### 2.4. Data Processing and DATA Analysis

#### Signal EMG

The pre-processing and analysis of the EMG data was performed on MATLAB software (The MathWorks Inc., Natick, MA, USA, version R2021b). To analyse EMG data, the signals were filtered using a 4th order low-pass Butterworth filter with a cut-off frequency of 5 Hz [44]. Then, the root mean square (*RMS*) of the EMG signal was calculated, which is the amplitude of the electrical signal generated by the muscle during the movement:(1)$$\mathrm{RMS}=\sqrt{\frac{1}{T}\cdot \int_{t-T/2}^{t+T/2}x{\left(t\right)}^{2}\cdot dt}$$
with *x(t*) the EMG signal to be analysed and *T* the signal time interval.

The *RMS* was then normalised to the subject’s MVC to reduce inter-individual variation [43], and to allow a comparison of the subject’s effort in relation to his maximum effort.

### 2.5. Statistical Analysis

For the statistical treatment, descriptive statistics were performed as well as comparisons of paired measurements using the JASP software (JASP Team, University of Amsterdam, Amsterdam, The Netherlands, version 0.16.3). Given the small sample size, a non-parametric statistical analysis has been performed using a Wilcoxon’s test to compare the means of the two conditions. The probability threshold for rejecting the null hypothesis has been established at 5% (i.e., *p*-value < 0.05) [45].

## 3. Results

### 3.1. Time Spent on Task

The time spent on the task in both conditions was not significantly different (*p*-value = 0.33) with a time spent of 399 ± 245 s with the exoskeleton and of 461 ± 333 s without the exoskeleton.

### 3.2. Muscle Activity during the Task

Regarding muscle activity during the task (see Table 1), there was a significant difference in the activity of the anterior deltoid with and without the exoskeleton (*p*-value = 2.62 × 10${}^{-2}$), with *RMS* values equal to 7.43±6.75% and 13.83±15.06% of MVC, respectively. Muscle activity of the latissimus dorsi was significantly higher in the “without exoskeleton” condition compared to the “with exoskeleton” condition (*p*-value = 4.7 × 10${}^{-2}$), with *RMS* values of 10.90 ± 7.16% and 9.49±7.38% of MVC, respectively.

No significant difference was observed with and without the exoskeleton for the biceps (*p*-value = 0.94), triceps brachii (*p*-value = 0.81), pectoralis major (*p*-value = 0.30) and longissimus (*p*-value = 0.44) (Figure 3).

The actual working time with and without an exoskeleton was not correlated with the muscle activity of the anterior deltoid (*p*-value = 0.44) and the latissimus dorsi (*p*-value = 0.5) during the tool cleaning task.

### 3.3. Perception of the Task

The perceived exertion with and without the exoskeleton was 2.43±2.49 and 3.57±2.29 respectively, although this difference was not significant (*p*-value = 0.26). Of all the subjects, five felt that the exertion with the exoskeleton was less than without the exoskeleton.

The perceived discomfort while wearing the exoskeleton was rated ‘low’, with a score of 2.88±2.57 out of 10 (Table 2). Of all the participants, two subjects reported significant discomfort while wearing the exoskeleton, with a score of 7/10.

The usability of the system, measured using the SUS, showed an average score of 69.69 out of 100 (69.69±14.04). The use of the exoskeleton may therefore be considered acceptable as a score of 70 is the threshold score for which the system is considered acceptable.

## 4. Discussion

The aim of this study was to assess the influence of an upper limb exoskeleton during a typical task performed by healthcare workers, i.e., a surgical tool cleaning task. This evaluation was obtained with objective (EMG) and subjective (questionnaires) data.

### 4.1. Flexor and Extensor Muscles of the Shoulder

Significant differences were observed in the anterior deltoid (*p*-value < 0.05) and latissimus dorsi (*p*-value < 0.05). These results are in line with the design of the Hapo MS^™^ passive exoskeleton as it was conceived to reduce the load on the upper limbs, particularly on the shoulders. Indeed, this exoskeleton assisted antepulsion movements which mainly involved the anterior deltoid. These results are in agreement with the literature, which showed that passive exoskeletons significantly reduced muscle load on the anterior deltoid muscle by 24 to 73%, depending on the study [3,16,24]. Thus, the passive exoskeleton Hapo MS^™^, seems to have a positive effect on the muscular solicitation at shoulder level during static tasks with low repetitive efforts. It may be suggested that a frequent use of this exoskeleton would decrease the rate of occurrence of MSDs around the shoulder. It was not possible to compare muscle solicitation of the latissimus dorsi as none of the previous studies analysed this muscle while wearing an exoskeleton. However, given the fact that this muscle is involved in several functional movements of the upper limbs, i.e., adduction, internal rotation and retropulsion of the upper arm, it may be suggested that the upper-limb exoskeleton used in the current study facilitates adduction or internal rotation of the arm. Current data about the latissimus dorsi may thus be used as a basis for future comparisons.

The other muscles of interest (biceps, triceps, pectoralis major) did not show any significant difference with and without exoskeleton, contrary to some studies. Indeed, in the study from Van Engelhoven et al. ([22]) where participants had to perform overhead assembly tasks, an increase in EMG amplitude of 80% was observed for the triceps brachii with a 20 Nm support. Theurel et al., 2018 ([3]) also observed a 97% and 107% increase in EMG amplitude during lifting (9 kg for men and 5 kg for women) and stacking tasks (15 kg for men and 8 kg for women). Other authors, such as Rashedi et al., 2014 ([16]) made the same observation with an increase of more than 30% in the EMG amplitude of the triceps brachii during overhead work with 2 handed weighted tool. However, Theurel et al., 2018 ([3]) observed no increase in EMG amplitude for the triceps brachii during carrying tasks (15 kg for men and 8 kg for women) but a decrease of 64%.

One possible explanation for this difference concerns the nature of the movement, as it allowed the exoskeleton to either assist the movement or to oppose the movement and thus increase the muscular solicitations. In the studies of Rashedi et al., 2014 ([16]) and Van Engelhoven et al., 2019 ([22]), participants were asked to perform an overhead task whereas in the study from Theurel et al., 2018 ([3]), participants had to lift and carry a box. More precisely, in the studies from Rashedi et al., 2014 ([16]) and Van Engelhoven et al., 2019 ([22]), the triceps brachii was an elbow extensor and was therefore assisted by the exoskeleton, while in the current study and in the study from Theurel et al., 2018 ([3]), the triceps brachii was a shoulder extensor and acted in opposition to the exoskeleton, which could result in greater muscle demands.

### 4.2. Longissimus Muscle Solicitation

During the task, the longissimus was the most solicited muscle, either with and without exoskeleton, with respective values of 45.83±20.16% and 46.80±16.79% of MVC. Indeed, the task of cleaning tools requires, among other things, significant lifting of the body, but also significant static moments during which the trunk and neck are flexed (see Figure 2). However, a passive upper limb exoskeleton has been employed through this study as the upper limbs were concerned with repetitive movements during the tool cleaning task. Indeed this task placed a high demand on the shoulders. Thus, the exoskeleton only slightly decreased the muscular load on the longissimus and this explains why the differences found on the longissimus with and without the exoskeleton are non-significant (*p*-value = 0.44). This result is in line with the study from Gillette and Stephenson, 2019 ([19]) who showed no influence of wearing an upper-limb exoskeleton on erector spinae muscle group.

However some authors observed a 25 to 81% increase in EMG amplitude of the erector spinae when wearing an exoskeleton compared to the condition without an exoskeleton [14,16]. A decrease of muscle solicitation have also been observed in the literature [18,20]. These results may be due to the nature of the task or to the type of exoskeleton which did not solicit the spinal erectors in the same way. Indeed, the studies from Alabdulkarim and Nussbaum, 2019 ([14]) and Rashedi et al., 2014 ([16]) were experimental laboratory studies which involved an overhead work task. This task may have been more demanding than the tasks described in the studies from Kim and Nussbaum, 2019 ([18]) and Gillette and Stephenson, 2018, 2019 ([19,20]), as it was field tasks of light assembly or cycle-job.

Furthermore, the exoskeletons used in the aforementioned studies were different and could explain the fact that there is no consensus concerning the influence of an upper-limb exoskeleton on erector spinae muscle group. Indeed, the study of Alabdulkarim and Nussbaum, 2019 ([14]) compared three different types of exoskeletons: the Fortis^™^ (Lockheed-Martin, Bethesda, MD, USA), the ShoulderX^™^ (SuitX, Emeryville, CA, USA) and the Fawcett Exovest^™^ (The Tiffen Company, Hauppauge, NY, USA). This study showed a higher muscle load on the erectors of the spine with the Fortis^™^, a full-body exoskeleton. Rashedi et al., 2014 ([16]) used the Fawcett Exovest^™^ (weight = 4.85 kg), a fairly heavy upper limb exoskeleton where the weight and torque are transfered through four pelvic pads and upper back pads [46], which may explain the higher load on the spinal erectors. The study from Kim and Nussbaum, 2019 ([18]) and the two studies from Gillette and Stephenson, 2018, 2019 ([20], 2019) assessed the Eksovest^™^(Ekso Bionics, San Rafael, CA, USA) exoskeleton and the Airframe^™^ (Levitate Technologies, San Diego, CA, USA), respectively. The EksoVest^™^ (weight = 4.3 kg) is an exoskeleton with a rigid structure. Through spring pressure on the structure, it can help to produce torque on the shoulder joint [46]. The Airframe^™^ has a similar system with two interlocking pulleys to adjust the spring tension, the assisting torque thus adapting to the elevation of the arm [46]. Thus, the exoskeletons used in the Rashedi et al., 2014 ([16]) and Alabdulkarim and Nussbaum, 2019 ([14]) studies were transferring weight and torque to another body area. In the studies from Kim and Nussbaum, 2019 ([18]) and Gillette and Stephenson, 2018, 2019 ([19,20]), the exoskeletons were assisting the movement by releasing the energy stored into the springs. In the current study, the Hapo MS^™^ exoskeleton worked with rigid rods on the same model than the Airframe^™^ and Eksovest^™^ exoskeletons. This may be a possible explanation for the non-significant decrease in longissimus muscle activity in our study.

### 4.3. Towards a Better Understanding and Integration of Exoskeleton at Work

Results from the current study are in agreement with results of field studies on the use of a passive exoskeleton for upper limb assistance. Indeed, a decrease of around 42% in muscle activation of the anterior deltoid has been observed with the exoskeleton. These results are comparable with the study from Wang et al., 2021 ([23]), who observed a decrease of 37.67–39.57% in anterior deltoid solicitation. Moreover, in most of the previous field studies, the results obtained were less significant than those from laboratory studies. Indeed, the study of Gillette and Stephenson, 2019 ([19]) observed a decrease in muscle activation of the anterior deltoid of 19%. Furthermore, De Bock et al., 2023 ([26]) compared muscle activation of different muscles with and without exoskeleton under laboratory and field conditions. They concluded that muscle activation was less important during field conditions and that the laboratory results should be interpreted with caution.

Concerning the subjective results obtained in the current study, only two subjects considered that wearing the exoskeleton was uncomfortable with a score of 7/10 for discomfort, whereas the group average was 2.88±2.57. These same two subjects also perceived exertion as difficult with an exertion perception score of 5/10 for ’subject 3’ and 7/10 for ’subject 7’. Subject 3’s discomfort and perceived exertion results were in line with his objective results. In fact, subject 3 was the only subject who obtained higher muscle loads for the pectoralis major and the longissimus, with respectively 91.3% and 70.6% higher solicitations. In addition, an increase of 157.1% in muscular activity of the biceps muscle was observed for this subject while wearing the exoskeleton compared to the situation without the exoskeleton. His other muscle groups showed a non-significant decrease of 9.8% for the triceps, 11.8% for the anterior deltoid and 12.5% for the latissimus dorsi. For subject 7, only the triceps showed a 39.1% increase with the exoskeleton compared to the condition without the exoskeleton.

The perception scores were not able to discriminate against the perceived discomfort or stress of the different muscle chains (forearm, arm, shoulder, back). It would therefore be interesting in a future study to propose to note the perceived effort as well as discomfort according to the different muscular chains, and not in general as it was done in the study from Alabdulkarim and Nussbaum [14].

The exoskeleton disposal was considered acceptable for three subjects out of eight, with a score higher than 70, while the other subjects gave a score between 55 and 67.5 out of 100. This may partly be due to the fact that the cleaning task lasted between 3 and 15 min and participants felt that the physical benefit was not significant enough compared to the time lost to allow the exoskeleton to be removed and stored. Thus, out of the seven participants, only one participant agreed with the frequent use of the exoskeleton. It has been previously showed that the acceptability of a system is part of a progressive and complex continuum [47]. To facilitate the integration of exoskeletons into the workplace, some applied researchers proposed a three-phase plan [48]. The first phase concerns the decision support, during which the evaluation of the physical workload is performed to further detail the most demanding tasks that could benefit from physical assistance. Characteristics of the exoskeleton have to be collectively validated, by involving workers as early as possible in the design and in the subsequent deployment of the exoskeleton. During the second phase, the selected exoskeleton has to be assessed on the basis of chosen criteria. Then, the exoskeleton has to be tested in the field with a learning period for workers, in order to become familiar with the exoskeleton. Following this second phase, the exoskeleton may be integrated into the company. The third and last phase, consists of collecting feedback about the use of the exoskeleton in the short, medium and long term. This may enable the exoskeleton to be adapted to evolving occupational situations. However, some phases may appear difficult to settle in some companies. This is particularly the case into hospitals, where workers are continuously replaced. Thus, it might be complex to evaluate long-term effects of an exoskeleton. Thus, field studies appear critical to evaluate exoskeletons in order to choose and deploy the most suitable exoskeleton for hospital staff.

Some limits may be associated with the protocol of this study. Firstly, even though it focused on one manual task, i.e., cleaning surgical instruments, with a similar mean duration, movement kinematics were not recorded with and without the exoskeleton. Although it would have complicated the experimental protocol, this would have also ensured that significant results between both conditions were only due to the influence of the exoskeleton. This might be achieved in a future studies by using on-body sensors integrating both inertial measurements and EMG in the same unit. Having more participants may have also supported this significance. However, all healthcare workers of the surgery unit have been recruited for this experiment. The fact that this study was conducted directly in the field considerably reduced the number of participants. This is a typical limitation related to ergonomic studies with an applied research perspective [49]. The main objective of this study was to provide valuable elements for the in-field assessment of an upper limb exoskeleton.

## 5. Conclusions

In conclusion, through this study, an upper limb exoskeleton was evaluated for a surgical tool cleaning task performed by healthcare workers. We showed that the longissimus muscle was mainly solicited during the task. It would therefore be interesting to reproduce this study with a back exoskeleton in order to evaluate the influence of this type of exoskeleton on the trunk and upper limb muscles. By reducing muscular loads on healthcare workers, work-related MSDs might be prevented in the long term.

The upper limb exoskeleton positively influenced anterior deltoid and latissimus dorsi during the task, while other muscles were not affected (triceps brachii, biceps brachii, brachioradialis, pectoralis major). The Hapo MS^™^ exoskeleton therefore appears to be suitable for this typical task of healthcare workers.

A perspective of this study might be the use of inertial measurement units placed on the worker in order to analyse kinematics while wearing the exoskeleton. Indeed, the use of an upper limb exoskeleton may have an influence on kinematics [50]. This parameter is thus important to consider when analysing the effects of the exoskeleton for the prevention of MSDs.

## Figures and Tables

**Figure 1 ijerph-20-03153-f001:**
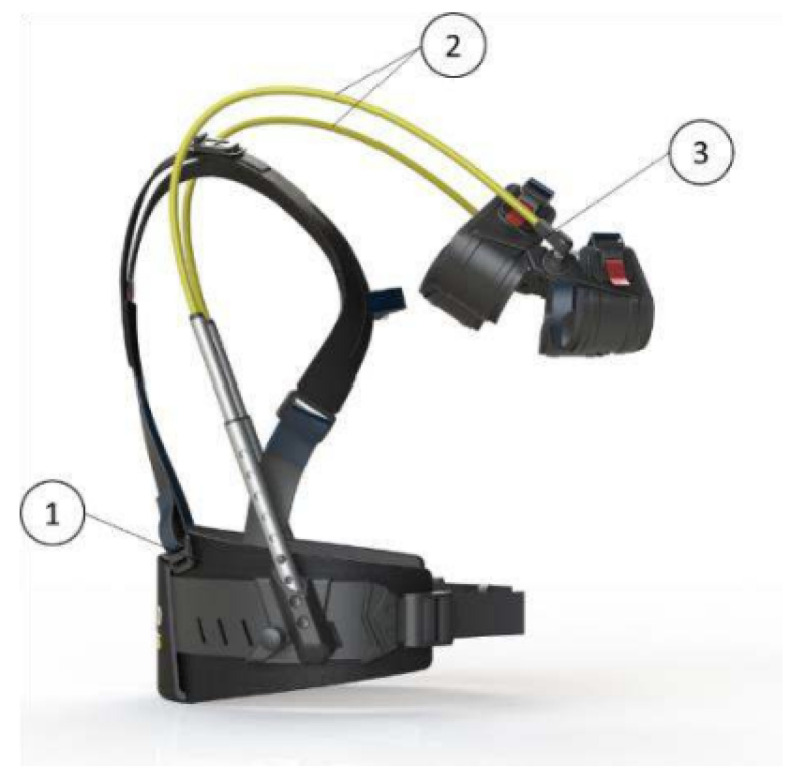
Description of Hapo MS^™^ exoskeleton model (Le Tellier et al., 2021 [37]). (1) harness; (2) springs; (3) double-interface.

**Figure 2 ijerph-20-03153-f002:**
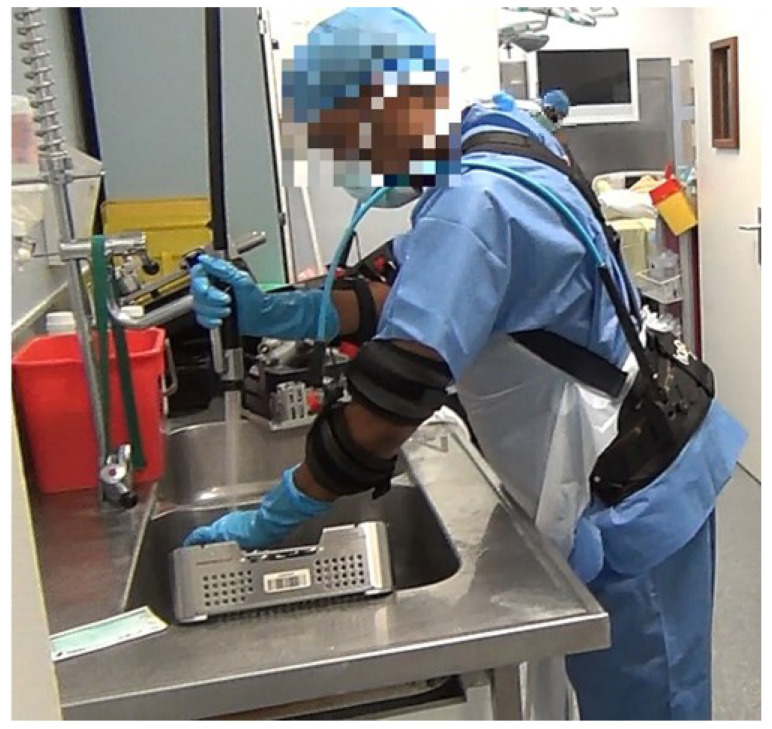
Healthcare worker wearing the exoskeleton during the task of cleaning surgical tools.

**Figure 3 ijerph-20-03153-f003:**
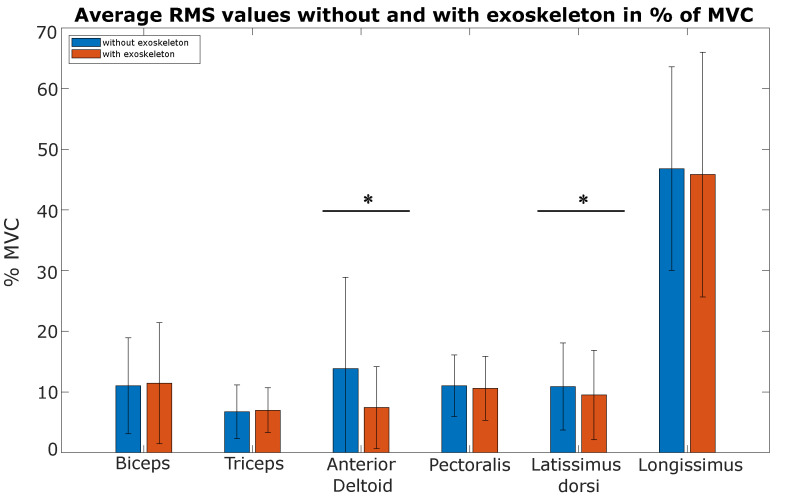
Comparison of mean *RMS* values between the two conditions (with and without exoskeleton) in pourcentage of MVC. Significant differences (*p*-value < 0.05) denoted by *.

**Table 1 ijerph-20-03153-t001:** Difference in muscle solicitation between the two conditions for all participants. The light green cells indicate a non-significant decrease in muscle activity with the exoskeleton, the dark green cells indicate a significant decrease, and the red cells show a non significant increase.

Subject	Biceps	Triceps	Anterior Deltoid	Pectoralis	Latissimus	Longissimus
1	14.3%	−23.4%	−58.2%	−5.1%	−5.3%	−0.5%
3	157.1%	−9.8%	−11.8%	91.3%	−12.5%	70.6%
4	−29.2%	−34.1%	−24.0%	−41.8%	−2.1%	−21.5%
5	−8.0%	52.9%	−37.9%	−9.3%	−40.2%	−7.2%
6	−41.8%	6.3%	−54.4%	−15.0%	−5.4%	−38.5%
7	−45.1%	39.1%	−52.1%	−12.8%	−37.5%	−0.2%
8	3.0%	59.2%	−22.4%	−16.5%	NaN	NaN
Group	4.1%	3.8%	−46.2%	−3.8%	−12.9%	−2.1%

**Table 2 ijerph-20-03153-t002:** Perception of the task (discomfort and effort), with and without exoskeleton, according to each subject.

Subject	Discomfort	Effort	
		Exo	Without Exo
1	1	3	7
3	7	5	0
4	1	1	3
5	1	0	3
6	0	0	2
7	7	7	5
8	3	1	5
Group (mean ± SD)	2.88 ± 2.57	2.43 ± 2.50	3.57 ± 2.13

## Data Availability

Not applicable.

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
