# Peer review of "Evaluation of a Passive Upper Limb Exoskeleton in Healthcare Workers during a Surgical Instrument Cleaning Task"

_ijerph, 2023, doi:10.3390/ijerph20043153_

Round 1

Reviewer 1 Report

The present study investigates the use of the HAPO exoskeleton to reduce upper limb muscles activations during a surgical tool cleaning task. The manuscript is well presented and written. However, I have several comments that need to be addressed. Especially, the question related to the duration and kinematics of the task and the one dealing with the participant number are very important.

General Comments:

A) The aim of the study could be better presented (see my specific comments). Moreover, isn’t driven by hypotheses?

B) The 2 conditions (with and without exo) are compared on the basis of 2 different movements, in terms of kinematics and duration. It would have been easier to replicate the first condition twice, to get the same movement between both conditions. By doing so, you would have ensured testing only the exo effect. Now that you have 2 different influences (exo, movement), how do you make sure that the observed results are only related to the influence of the exoskeleton?

C) Given the variability of the task and the need for instance to discuss participant 3 and 7 separately, 8 participants might not be enough to provide robust results. This should at least be seen as a limitation of the study.  

Specific Comments:

Introduction:

- p1 line 33: what do you mean by by physical tasks such as repetitive movements or lifting tasks? Could you please give some examples?

- p2 line 66: reference 31 already investigated the use of an exoskeleton. What does the present study try to add to the body of knowledge with respect to this previous one?

- p2 line 68: is a surgical tool cleaning task subject to high loadings and/or repetition?

Methods:

- p3 line 90: this is not a typical stiffness unit. What is meant with kg when expressing stiffness?

- p3 line 94: I don’t understand. Why isn’t the assistance selected wrt. the mass of the manipulated tool? Is the level of assistance also recommended according to the range of motion?

- p3 line 100: why not the posterior deltoid if you are looking at antagonist’s activation? Is the dorsalis major the latissimus dorsi? Where do you get the longissimus at the thorax level with surface EMG? The latter muscle isn’t an antagonist? Wouldn’t the trapezius medius be relevant?

- p4 line 127-28: I don’t understand. Is there a possibility that both conditions (with and without exo) were conducted over different movements and durations?

- p4 line 133-34: please check wording?

- p4 line 139: low pass at 5Hz helps to create an envelope, what is unnecessary here. The reference you quote here rather argue about high pass filter and estimation of force, which is not your purpose here. I don’t understand this EMG signal treatment?

- p5 line 149: rather than the number of subjects, this decision could be made on the data distribution. Don’t data follow a normal distribution?

Results:

- p5 line 155: I think seconds should be presented with 3 digits (399 vs. 245).

- p5 line 160-62: please provide the exact p value. I don’t understand how the dorsalis major can by significant with such SD values?

- p5 line 169-71: what is meant by this analysis? What is the rationale?

- Figure 3: what does * mean? The * for pectoralis is wrong and should be attributed to the dorsalis.

- Table 1: Isn’t mentioned in the text. Should keep the same muscles order than in Figure 3. EMG data for participant 8 have been partially used. This should be stated in the methods.  

- p6 line 178: ‘as low’? Wording?

- p6 line 179: ‘only’ is not appropriate as it represents 25%

Discussion:

- p6 line 194: do you mean Figure 2?

- p7 line 201-232: I don’t understand the purpose of this argumentation. As these studies didn’t work on the same task, there is no point to compare them. The lack of significant results for the longissimus is simply the fact that the present exoskeleton doesn’t target these muscles, as you have stated. If significances would have been found, this could have been discussed in details, but this is not the case.

- p7 line 235: you should start your discussion with this result as it is the primary one.

- p7 line 239-30: I don’t think the dorsalis major supports the retropulsion here. It may rather help during adduction and internal rotation of the arm? Why would an exoskeleton support 2 antagonist actions, i.e. ante and retropulsion? Couldn’t the reduced dorsalis major activity be related to another function of the HAPO: increased adduction or internal rotation?

- p7 line 295: 157% for the biceps muscle

Author Response

We would like to thank both reviewers for their thorough review as we are convinced it improved the quality of the article. We replied to each of the comments in the document (responses in red). Corresponding modifications in the manuscript have also been highlighted in red.

Reviewer 2 Report

This paper deals with the evaluation of upper limb exoskeleton for a surgical tool cleaning tasks performed by healthcare workers. The analysis of the exoskeleton was carried out in subjective and objective methods, e.g. EMG measuring and questionnaires. The results showed that the exoskeleton was effective for anterior deltoid and major dorsalis. The experimental design is appropriate. But the limit is that the number of subject N is too small. My question is as follows.

1)     The evaluation was done with seven healthcare works. And they performed cleaning task with and without exoskeleton in random order.  The height was 178.88+/-5.34 cm, which means the upper limb exoskeleton does not fit every subject perfectly. Is there any correlation between height or age with muscle solicitation (in table 1) or discomfort (in table 2) ?

2)     Regarding the EMG, how about peak value in comparison of with and without exoskeleton?

Author Response

(The authors gave the same response as above.)

Round 2

Reviewer 1 Report

The authors have clearly addressed my previous comments. I believe this has helped to improve the quality of the manuscript. Nice work.

I have a final comment: be consistent with the use of latissimus (table and figure) or dorsalis (text).

Author Response

We thank the reviewer for his time. The terms 'latissimus dorsi' has now been used in the whole manuscript.

Moreover, an English native speaker has read and corrected the document.

Best regards,

Reviewer 2 Report

The manuscript has been revised according to the reviewers' comments. It is enough to be published in IJERPH.

Author Response

We thank the reviewer for his time. An English native speaker has read and corrected the document.

Regards,